# Spectral tuning and deactivation kinetics of marine mammal melanopsins

Jeffry I. Fasick[1]*, Haya Algrain[2], Courtland Samuels[3], Padmanabhan Mahadevan[1], Lorian E. Schweikert[4], Zaid J. Naffaa[5], Phyllis R. Robinson[2]

1 Department of Biological Sciences, The University of Tampa, Tampa, Florida, United States of America, 2 Department of Biological Sciences, University of Maryland Baltimore County, Baltimore, Maryland, United States of America, 3 Department of Chemistry, University of South Florida, Tampa, Florida, United States of America, 4 Department of Biology and Marine Biology, University of North Carolina Wilmington, Wilmington, North Carolina, United States of America, 5 Department of Biological Sciences, Kean University, Union, New Jersey, United States of America

* jfasick@ut.edu

**Data Availability Statement:** The sequences underlying this study have been uploaded to GenBank. The accession numbers are located in the Supporting Information files.

## Abstract

In mammals, the photopigment melanopsin (Opn4) is found in a subset of retinal ganglion cells that serve light detection for circadian photoentrainment and pupil constriction (i.e., mydriasis). For a given species, the efficiency of photoentrainment and length of time that mydriasis occurs is determined by the spectral sensitivity and deactivation kinetics of melanopsin, respectively, and to date, neither of these properties have been described in marine mammals. Previous work has indicated that the absorbance maxima ($\lambda_{max}$) of marine mammal rhodopsins (Rh1) have diversified to match the available light spectra at foraging depths. However, similar to the melanopsin $\lambda_{max}$ of terrestrial mammals (~480 nm), the melanopsins of marine mammals may be conserved, with $\lambda_{max}$ values tuned to the spectrum of solar irradiance at the water's surface. Here, we investigated the Opn4 pigments of 17 marine mammal species inhabiting diverse photic environments including the Infraorder Cetacea, as well as the Orders Sirenia and Carnivora. Both genomic and cDNA sequences were used to deduce amino acid sequences to identify substitutions most likely involved in spectral tuning and deactivation kinetics of the Opn4 pigments. Our results show that there appears to be no amino acid substitutions in marine mammal Opn4 opsins that would result in any significant change in $\lambda_{max}$ values relative to their terrestrial counterparts. We also found some marine mammal species to lack several phosphorylation sites in the carboxyl terminal domain of their Opn4 pigments that result in significantly slower deactivation kinetics, and thus longer mydriasis, compared to terrestrial controls. This finding was restricted to cetacean species previously found to lack cone photoreceptor opsins, a condition known as rod monochromacy. These results suggest that the rod monochromat whales rely on extended pupillary constriction to prevent photobleaching of the highly photosensitive all-rod retina when moving between photopic and scotopic conditions.

**Funding:** Funding for this work was provided by a Research Innovation and Scholarly Excellence Award/Dana Foundation Grant (J.I.F.) and a Biology Summer Research Fellowship from The University of Tampa. P.R.R. was supported by National Science Foundation grant R01 EY027202-02. The funders had no role in study design, data collection and analysis, decision to publish, or preparation of the manuscript.

**Competing interests:** The authors have declared that no competing interests exist.

## Introduction

The vertebrate visual system evolved to detect light for both image forming vision and non-image forming processes, and emerged approximately 500 million years ago [1]. The ensuing nocturnal bottleneck in mammalian evolution influenced the evolution of mammalian vision with the loss of at least 3 classes of opsins (SWS2, Rh2 and Opn4x) [2, 3]. Today, the majority of mammals possess a long-wavelength sensitive (LWS) and short-wavelength sensitive (SWS1) cone photoreceptor class; one rod (Rh1) photoreceptor class; and one melanopsin retinal pigment (Opn4m) which is expressed in intrinsically photosensitive retinal ganglion cells (ipRGCs). The radiation of mammals into specialized niches, however, has resulted in further diversification of the mammalian visual system including cone trichromacy occurring in humans and other primates [4, 5] as well as rod monochromacy occurring in xenarthrans and some cetacean species [6–8].

Marine mammals have undergone a variety of adaptations to their visual systems upon their return to the sea. These mammals, which include the orders Cetacea, Sirenia, and Carnivora which includes pinnipeds (Otariidae, Phocidae, Odobenidae), polar bears (Ursidae) and sea otters (Mustelidae), possess eyes that, for most species, have been modified to enhance image formation underwater. Likewise, many marine mammal species possess rod and cone photoreceptors that contain visual pigments that are spectrally tuned to align to the available underwater spectral radiance. Although most marine mammals possess both functional rod and LWS cone photoreceptors, only Sirenia, Ursidae and Mustelidae possess both LWS and SWS1 cone photoreceptors allowing for dichromatic color vision [9, 10]. All cetacean and pinniped species have lost functional SWS1 cone photoreceptors resulting in the loss of typical dichromatic color vision [3, 7, 8, 11–17]. Almost all baleen and beaked whale species have also lost the LWS cone visual pigment resulting in rod monochromacy [7] while retaining the LWS cone soma and maintenance of rod/cone based retinal circuitry [8].

A small subset of retinal ganglion cells, ipRGCs, in the mammalian retina express the photopigment melanopsin (Opn4) and represent a third class of photoreceptor that predominantly mediates non-image forming visual functions. In terrestrial mammals, ipRGCs provide photic information for a number of light-dependent processes, including circadian photoentrainment, pupil constriction, suppression of pineal melatonin, and direct regulation of sleep, mood, and learning [18–27]. Recently, it has been shown that ipRGCs are also involved in image forming vision where they contribute to contrast sensitivity [28–34]. ipRGCs differ from the classical rod and cone photoreceptors in both physiology and the light-activated biochemical cascade. Since the discovery of melanopsin-expressing ipRGCs over 20 years ago [35, 36], these neurons and their expressed pigment have been the subject of intense research primarily in terrestrial dichromatic mammals, such as rodents, with significant advances being made in elucidating the anatomy and functions of ipRGCs as well as the rod/cone input to these cells [22, 24, 37, 38]. Vertebrate melanopsins exhibit a higher sequence homology to invertebrate rhabdomeric (R-type) visual pigments than to mammalian cone and rod visual pigments (C-opsins) [35, 36], while ipRGCs relay radiance information from the retina to brain regions that regulate the light-dependent processes mentioned above. Previously thought to be a functionally uniform population, ipRGCs in the mouse retina are now known to include six different subtypes (M1-M6) that can be differentiated based on morphological and electrophysiological criteria, projection targets, and function [39].

Presently, there is a major gap in our knowledge of the role of ipRGCs and their expressed pigment in mammals that have adapted to aquatic environments, including cetacean species that display rod monochromacy. Here, we address two questions pertaining to marine mammal melanopsins as well as to the functional roles of ipRGCs in aquatic rod monochromats: 1)

have marine mammal melanopsins diverged from terrestrial melanopsins with regards to their spectral sensitivities? and 2) do ipRGCs and their expressed pigment from cetacean rod monochromats play a role in protecting rod photoreceptors from photobleaching by altering the kinetics of the pupillary light reflex (PLR)?

The first question is based on the observation that marine mammal visual pigments are spectrally tuned to overlap the underwater spectral radiance associated with foraging depth [9, 12, 40–49]. Modulating the spectral sensitivity results in relatively large, blue-shifted absorbance spectra from visual pigments of deep-diving pelagic marine mammals, and relatively slightly red-shifted absorbance spectra from the visual pigments of near-coastal and riverine species, when compared to their terrestrial counterparts. Unlike terrestrial mammals, most marine mammal species reside in two spectrally distinct photic environments: the surface where they breathe and utilize broadband light spectra, and at foraging depth where they can utilize narrowband light spectra. The spectral irradiance at foraging depth is typically blue-shifted from the 500 nm region of the visible light spectrum where terrestrial Rh1 pigments maximally absorb [9, 40–51]. Melanopsin absorbance maxima from a variety of vertebrates, including mouse and human, appear to be constant, around 480 nm [52–56]. Because all marine mammals spend a significant portion of time at the surface of the water to breathe, we hypothesized that marine mammal melanopsins will be uniformly tuned to the spectrum of solar irradiance as opposed to being tuned to spectral variations found in their aquatic environments. Accordingly, we hypothesized that marine mammal melanopsins possess few if any amino acid substitutions that would result in a deviation from the absorbance maxima of typical terrestrial mammalian melanopsins (~480 nm). To test this hypothesis, we aligned deduced amino acid sequences from 17 marine mammal *Opn4* gene sequences across three Orders and identified all non-conservative amino acid substitutions within the Opn4 transmembrane domains including residues forming the chromophore binding pocket. After homology modeling of the marine mammal Opn4 sequences, relative measurements were made from the functional group of each amino acid substitution to either the Schiff base lysine or the β-ionone ring of the chromophore to predict relative blue- or red-shifts of the modeled pigments based on previous studies [57–59].

The second question pertains to the PLR and the role of Opn4-expressing ipRGCs in cetacean rod monochromats with regards to this function. The PLR is the constriction and recovery of the pupil in response to light and is critical in reducing photoreceptor bleaching while allowing for fast dark adaptation [60]. Rods and ipRGCs influence the PLR over a wide range of irradiance levels [60–62]. Although rod photoreceptors are capable of controlling the PLR at relatively high irradiance levels with an upper limit of 10–12 log photons·cm$^{-2}$·s$^{-1}$ [60], when irradiance levels are above these limits, the PLR is controlled by ipRGCs expressing melanopsin [61]. Upon light activation under scotopic conditions, rod photoreceptors direct a rapid onset of ipRGC firing, and thus pupil constriction, after a delay of only 150 ms [60–62]. Under photopic conditions, however, the relatively slow activation kinetics of melanopsins expressed in ipRGC's results in maximum pupil constriction after approximately 30s [63–66] which most likely would result in significant, if not complete, rod photobleaching before full pupil constriction in rod monochromats. In this instance, the rod monochromat would be rendered blind in bright light conditions until significant reconstitution of chromophore is completed. To counter photoreceptor bleaching in bright light conditions, mammalian rod monochromats may have evolved and selected for rod visual pigments that reconstitute chromophore significantly faster or a compensatory mechanism to maintain an extended PLR wherein the pupil dilates considerably slower. With regards to the latter countermeasure, we hypothesized that cetacean rod monochromats have adapted by altering the phosphorylation sites of the melanopsin carboxyl tail in a manner that results in prolonged deactivation, and thus

prolonged pupil constriction. To test this hypothesis, we compared the putative phosphorylation sites found on the Opn4 carboxyl terminal domain from cetacean rod monochromats with those from marine mammals that have retained duplex retinae. Subsequently, calcium imaging assays were performed to determine and compare the deactivation rates of melanopsin from these species.

## Materials and methods

### Marine mammal melanopsin (*Opn4*) sequences

Domestic cow *Opn4* (*Bos Taurus*, GenBank accession no. NM_001192399) was used as a query sequence in nucleotide blast analyses to identify orthologs from other Cetartiodactyla marine mammal genomes including minke whale (*Balaenoptera acutorostrata*, XM_007174566), sperm whale (*Physeter microcephalus*, XM_007120485), bottlenose dolphin (Tursiops truncatus, XM_019925814), Pacific white-sided dolphin (*Lagenorhynchus obliquidens*, XM_027129620), killer whale (Orcinus orca, XM_004273009), beluga whale (Delphinapterus leucas, XM_022562447), Yangtze finless porpoise (Neophocaena asiaeorientalis asiaeorientalis, XM_024731433), and Yangtze river dolphin (*Lipotes vexillifer*, XM_007458352). *B. taurus Opn4* was also used to identify orthologs from Weddell seal (*Leptonychotes weddellii*, XM_006740386), walrus (*Odobenus rosmarus divergens*, XM_004410203), West Indian manatee (*Trichechus manatus latirostris*, XM_023740472), and polar bear (*Ursus maritimus*, XP_008694349). The mouse (*Mus musculus*) *Opn4* long-form (EDL24885) was included for comparison. Blasts were optimized for either highly similar or somewhat similar sequences using default settings including word size (11), match/mismatch scores (2/-3) and gap costs (existence: 5; extension: 2). Bowhead whale (*Balaena mysticetus*) *Opn4* was identified by blast analysis using *B. taurus Opn4* to search the Bowhead Whale Genome Resource database [67]. Eyes from the following species were recovered from dead stranded animals and used to PCR amplify reversed transcribed retinal cDNA using oligonucleotide primers designed to amplify partial *Opn4* sequences: North Atlantic right whale (*Eubalaena glacialis*, GenBank accession no. OK_169905), humpback whale (*Megaptera novaeangliae*, OK_169906), dwarf sperm whale (*Kogia sima*, OK_169907), harbor porpoise (*Phocoena phocoena*, OK_169908) and bottlenose dolphin (*Tursiops truncatus*). Maintenance of frozen tissues was approved by Kean University's IRB. Total RNA was isolated using the RiboPure Kit (Applied Biosystems/Ambion, Austin, TX) following the manufacturer's instructions for tissue sample preparations. First strand cDNA was generated from total RNA using an oligo $dT_{18}$ primer and reverse transcriptase provided in the Reverse Transcription System kit (Promega, Madison, WI) following the manufacturer's instructions. Coding regions of *Opn4* cDNAs were PCR amplified with 1μM of respective forward and reverse primers (S1 Table) using Amplitaq Gold master mix (Invitrogen, Carlsbad, CA), with PCR cycle parameters being found in S2 Table. PCR products were sequenced directly using both forward and reverse primers by Eurofins Genomics (Louisville, KY). *Opn4* cDNA sequences were based on a minimum consensus double stranded sequence from two independent PCR amplifications.

### Sequence analysis: Structural modeling and molecular evolutionary analyses

Phylogenetic trees were generated after sequence alignments using a Bayesian Inference method with a Metropolis Markov chain Monte Carlo method using MrBayes in Geneious Prime 2019 using the following settings: HKY85 substitution model; a gamma-distributed rate of variation across all sites with a gamma category value of 4; chain length of 1,100,000

generations with subsampling frequency of 200, burn-in length of 100,000, 4 heated chains with 14,283 random seeds. The resulting trees were visualized by FigTree v1.4.4 [68].

Marine mammal *Opn4* deduced amino acid sequences were used for homology-based three-dimensional structural modeling using LOMETS software [69]. Top ranked LOMETS structures were imported into PyMOL software (The PyMOL Molecular Graphics System, Version 2.0 Schrödinger, LLC) and aligned with squid Rh1 (PDB ID 2Z73; [70]) to estimate the distance from amino acid to the protonated Schiff base nitrogen and the β-ionone ring of the chromophore.

Marine mammal *Opn4* nucleotide sequences were codon aligned using the RevTrans2.0b server with default options [71]. The codon alignment was then subjected to positive and negative selection analyses using the Selecton server [72]. The Mechanistic Empirical Combination (MEC) model was used in Selecton with the default JTT amino acid matrix.

## *Opn4* constructs

Full-length *Opn4* coding sequences for West Indian manatee (*Trichechus manatus*, XM_004386107) and bowhead whale (*Balaena mysticetus*, bmy_16888) were obtained from GenBank and Bowhead Whale Genome Resource databases respectively. Synthetic *Opn4* cassettes were designed with 5' EcoRI and 3' NotI restriction sites. The coding sequence of the 1D4 epitope was added to the carboxy-terminal domain (CTD) sequences immediately 5'of, and in-frame with, the stop codon. The coding sequence for the 1D4 epitope tag (last eight amino acids of bovine rhodopsin, TETSQVAPA) was added to facilitate immunodetection of all constructs synthesized. The *Opn4* sequences were human codon optimized for expression in HEK293 cells (Integrated DNA Technologies, Coralville, IA). Constructs were then synthesized as gBlocks (Integrated DNA Technologies). West Indian manatee and bowhead whale *Opn4* constructs were PCR amplified with 10μM of respective forward and reverse primers (S1 Table) using AccuPrime PFX DNA polymerase (Thermofisher, Waltham, MA). PCR cycle parameters for West Indian manatee and bowhead whale are found in S3 Table. PCR products were purified using NucleoSpin Gel and PCR Clean-up kit (Macherey-Negel, Bethlehem, PA) and double digested with EcoRI and NotI. Digested PCR products and vector were ligated in a 3:1 insert to vector ratio (reaction volume 40 μl) using T4 DNA Ligase (Promega, Madison, WI), incubated overnight at 4˚C, and transformed into chemically competent *E. coli*. Ampicillin resistant colonies were picked and cultured in LB$^{Amp}$ broth. Plasmid DNA was purified using the Nucleobond Xtra Midi kit (Macherey-Negel, Bethlehem, PA). Coding domain sequences were confirmed in both directions (Genewiz, Inc., South Plainfield, NJ) using plasmid specific primers *via* sanger sequencing. Mouse *Opn4* (NM_001128599.1) appended with the 1D4 epitope and subcloned into pMT3 [73] was used as a control. Cloning of the C-phosphonull mouse Opn4 construct was previously described [63].

## Signaling kinetics of *Opn4* constructs using a fluorescent calcium imaging assay

Transfected cells were harvested 24 h post-transfection and reseeded into a 96-well plate (Corning, Corning, NY) at a density of $1 \times 10^5$ cells per well and dark adapted for 18–24 hrs in 5% $CO_2$ at 37˚C. Forty-eight h post-transfection, cells were incubated with 20 mM 9-*cis*-retinal (Sigma-Aldrich, St. Louis, MO) in the presence of the fluorescent calcium indicator, Fluo-4 AM (Invitrogen) containing 5mM probenecid (Invitrogen) for 1 h in the dark. Fluorescence measurements were taken at an excitation of 485 nm and emission of 525 nm every second for 300 s after a flash exposure (40Hz) on a Tecan Infinite M200 microplate reader (Tecan, Männedorf, CH).

## Statistical analysis of Opn4 deactivation rates

Data across four transfections (8 replicates per transfection) were normalized, averaged and the deactivation rates were calculated for 300s assay runs. The deactivation rate of signaling kinetics corresponds to data measurements taken after the peak of fluorescence from calcium imaging assay data. The deactivation rate was fitted to a one phase decay function $[Y = (y0—plateau) \cdot exp(-k \cdot x) + plateau]$, where y0 is the fluorescence value at t = 0, plateau is the lowest deactivation (y) value, exp is the decay constant (2.71828), k is the rate constant and x is time in seconds. Statistical significance was calculated by an unpaired t-test of Opn4 constructs with respect to mouse Opn4 control. Error bars represent standard error of the mean. All data were analyzed and plotted using GraphPad Prism software (GraphPad Software Inc., Sand Diego, CA).

## Skyward solar spectral radiance measurements at dawn

On August 6, 2015, spectral radiance measurements ($\mu W \ cm^{-2} \ nm^{-1} \ sr^{-1}$) were taken from a vessel anchored in Florida Bay approximately 10 miles north of Marathon, FL (GPS Location: 24.84; -81.04) with weather conditions of 40% cloud cover and a last quarter moon (57% visible). Measurements were made with a Hyper OCR Hyperspectral Radiometer (Satlantic, Halifax, NS) using SatView and Prosoft software supplied by the manufacturer. The radiometer was oriented approximately 60˚ to vertical at compass point 300˚ to avoid direct moonlight. Measurements were taken continually, at three second intervals, beginning at astronomical twilight (05:35 EDT, solar elevation: -4.87˚) and continuing through nautical twilight (06:05 EDT, solar elevation: 1.8˚), civil twilight (06:33 EDT, solar elevation: 7.69˚) and finishing 15 min after sunrise (06:58 EDT) or approximately 07:15 EDT (solar elevation: 16.91˚). Radiance spectra were plotted at 10 nm intervals, normalized by the integral (i.e., area under the curve), and scaled from zero to one. Presenting environmental light spectra as weighted by the integral, also known as spectral form, best represents the distribution of photons in a given spectrum, making the data less sensitive to the issues of binning the data by light wavelength or frequency [74]. A hypothetical $A_1$-chromophore-based Opn4 spectrum with absorbance maximum of 480 nm was generated by shifting the absorbance data of bovine rhodopsin ($\lambda_{max}$ = 500 nm) by -20 nm.

# Results

## Alignment, phylogeny, and spectral tuning of marine mammal melanopsins

Amino acid sequences spanning the transmembrane domains (TMDs) from 17 marine mammal *Opn4* coding regions are shown in Fig 1 and include members from Infraorder Cetacea, as well as the Orders Sirenia and Carnivora. The overall architecture of the marine mammal melanopsins examined is similar to those of previously reported mammalian melanopsins with conserved residues including Lys337 (mouse *Opn4* numbering), the site of retinal linkage [75, 76]; Glu214, proposed protonated Schiff base (PSB) counterion [77–79]; as well as conserved cysteines and asparagines at sites of disulfide bridge linkage [35] and glycosylation, respectively. To determine the evolutionary relationships of marine mammal melanopsins, we inferred the phylogeny of marine mammal *Opn4* sequences with other vertebrate opsin genes using a Bayesian analysis to generate the consensus tree shown in Fig 2. The phylogenetic relationships of the marine mammal *Opn4* sequences shown in Fig 2 are consistent with the basic taxonomic relationships between each Order as well as within the infraorder Cetacea. Closer examination of the *Opn4* sequences shows two clades containing sequences from either the

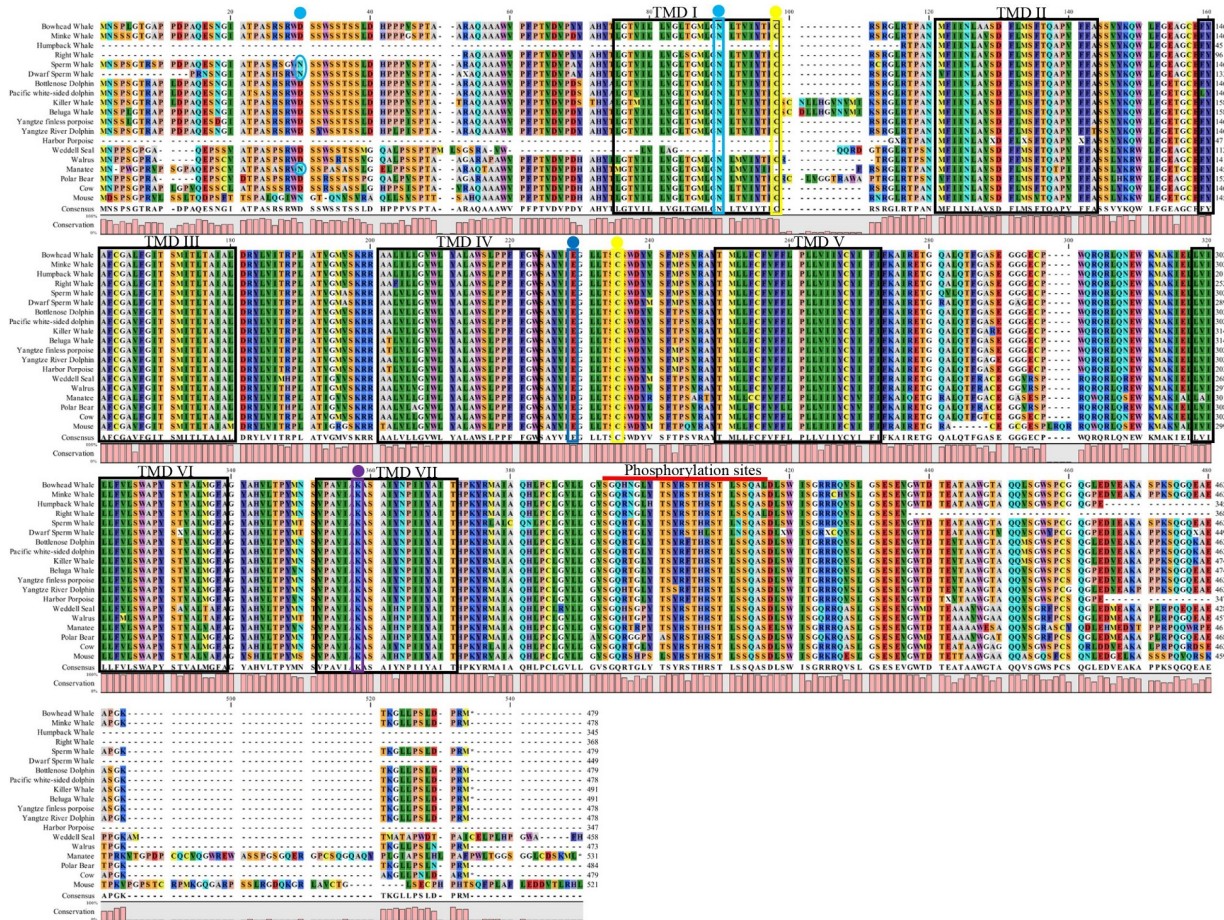

**Fig 1. Alignment of marine mammal melanopsins (*Opn4*).** Transmembrane domains (TMDs; boxed in black) were predicted using TMHMM Server Version 2.0 (http://www.cbs.dtu.dk/services/TMHMM) using the mouse Opn4 sequence. Conserved residues are boxed or circled and include: (1) glycosylation sites (asparagine-N) in the amino terminal domain and TMD I (predicted using NetGlyc Server 1.0 (www.cbs.dtu.dk/services/NetNGlyc) are marked with teal dots and are circled or boxed; (2) conserved cysteines (C) at positions 95 and 220 involved in disulfide bond formation are marked with yellow dots and are boxed; (3) proposed counterion residue, glutamate (E) at position 214, is marked with a blue dot and are boxed; (4) lysine (K) at position 337 that links to the chromophore is marked with a purple dot and is boxed; (5) potential serine (S) and threonine (T) phosphorylation sites spanning positions 372–395 in the carboxyl terminal domain are highlighted with a red bar. Mouse *Opn4* was used for amino acid numbering with cow *Opn4* used for comparison to a terrestrial Cetartiodactyla species.

mammalian-like (m) *Opn4* class or the xenopus-like (x) *Opn4* class, with all marine mammal *Opn4* sequences being grouped in the m-like class.

Potential spectral tuning positions were first identified by manual examination of amino acid alignments, as shown in Fig 1, followed by homology modeling and distance measurements from the functional group of each amino acid substitution to either the Schiff base lysine (Lys337 in mouse) or the β-ionone ring of the chromophore. Non-conserved residues occurring within the seven transmembrane domains that differed from mouse Opn4 ($\lambda_{max}$ = 480 nm; [52]) were examined in order to predict relative blue- or red-shifts of the homology modeled pigments. As shown in Fig 3, a total of ten nonconserved substitutions across all species were identified within the Opn4 transmembrane domains which could possibly influence the melanopsin absorbance spectrum for several marine mammal species. Two nonconserved substitutions were found in transmembrane domain I (TMI): Thr89Met, with walrus, manatee and polar bear all possessing methionine at this position; and Thr93Iso, with manatee

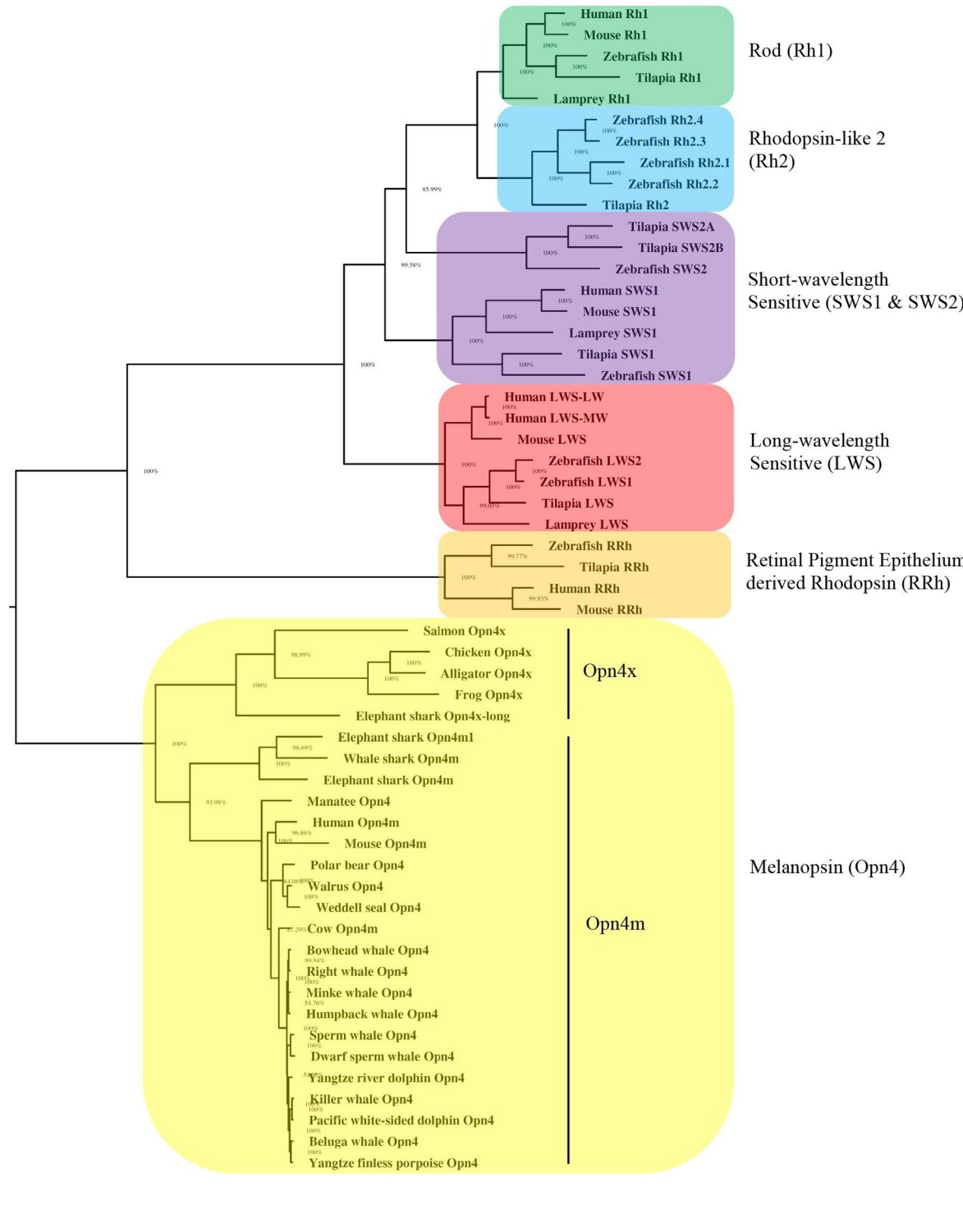

**Fig 2. Phylogeny of marine mammal melanopsins (*Opn4*).** A Bayesian inference method performed with Markov chain Monte Carlo method to estimate the posterior distribution of model parameters (represented as percentages) at the base of each node. The tree shows the relative position of vertebrate rod (*Rh1*), rhodopsin-like 2 (*Rh2*), short-wavelength sensitive (*SWS1* and *SWS2*) cone, long-wavelength sensitive (*LWS*) cone, retinal pigment epithelium-derived rhodopsin (*RRh*), and melanopsin (*Opn4*). Each opsin class is colored by clade: *Rh1* (green); *Rh2* (blue); *SWS* (violet); *LWS* (red); *RRh* (orange); and *Opn4* (yellow). Scale bar indicates the number of nucleotide substitutions per site. Accession numbers for the marine mammal *Opn4* sequences are found in Materials and methods and S3 Table.

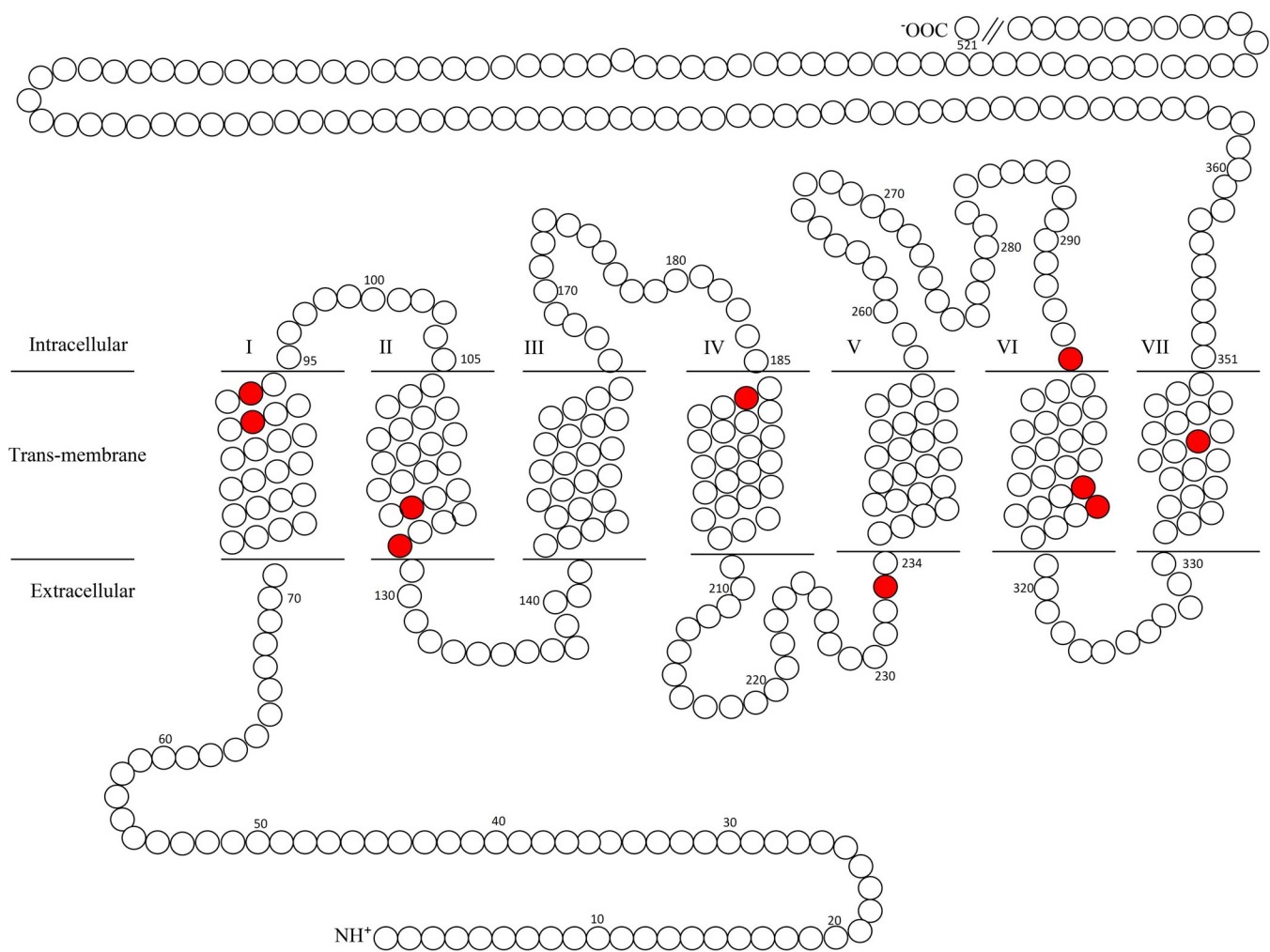

**Fig 3. Nonconservative amino acid substitutions between marine mammals, cow, and mouse melanopsins (Opn4) within transmembrane domains.**
Nonconservative amino acids substitutions (red; consensus residue: Position: Substitution) are as follows: Transmembrane region (TM) I: Thr89Met (walrus, manatee, polar bear); Thr93Iso (manatee); TMII: Ala123Thr (manatee); Ala128Thr (Yangtze river dolphin); TMIV: Ala187Thr (beluga whale, harbor porpoise, Yangtze finless porpoise,); ECL II: Ala233Thr (manatee); ICL III: Glu295Ala (manatee and mouse); TMVI: Thr311Ala (Weddell seal); Val/Met315Thr (Weddell seal, walrus); TMVII: Tyr342His (Weddell seal, walrus, manatee, polar bear and mouse). Opn4 numbering is from mouse.

possessing isoleucine at this position. Two nonconserved substitutions were found in TMII: Ala123Thr, with manatee possessing threonine at this position; and Ala128Thr with Yangtze river dolphin possessing threonine at this position. A single nonconserved substitution was found in TMIV: Ala187T with harbor porpoise, beluga whale, and Yangtze finless porpoise possessing threonine at this position. Although TMV lacked any nonconserved amino acid substitutions, a single substitution (Ala233Thr) was identified in extracellular loop (ECL) II immediately adjacent to TMV with manatee possessing threonine. Three nonconserved substitutions were found in or very near TMVI: Glu295Ala, with both manatee and mouse possessing alanine at this position in intracellular loop (ICL) III; Thr311Ala with only Weddell seal possessing alanine at this position; and Val/Met315Thr with both Weddell seal and walrus possessing threonine at this position. A single nonconserved amino acid substitution was identified in TMVII: Tyr342His with Weddell seal, walrus, manatee, polar bear, and mouse all possessing histidine at this position.

Potential spectral tuning residues most likely influence the electrostatic interaction between the PSB and counterion at the site of chromophore attachment, or along the polyene chain and the β-ionone ring structure of the chromophore. Homology modeling to squid rhodopsin was used to structurally model the mammalian Opn4 amino acid sequences to predict the influence that these non-conserved amino acid substitutions have on spectral tuning. The results shown in Table 1 identify the amino acid substitutions at the ten positions shown in Fig 3 along with measurements from the amino acid functional groups to either the PSB or the β-ionone ring (or both) using squid Rh1 as template. Of the ten Opn4 amino acid positions shown in Fig 3 and listed in Table 1, only four marine mammal amino acid substitutions are positioned in such a way that the respective functional groups are proximal to the chromophore binding pocket and are within ~10 Å to the chromophore when modeled using squid Rh1 as a template (Fig 4). When the R-groups for each Opn4 transmembrane substitution described above were modeled and measured to the Schiff base lysine terminal NZ atom or the closest atom of the β-ionone ring, the only substitutions with distances < ~10Å are found in manatee (Ala123Thr, 9.5Å to PSB; predicted blue-shift); Yangtze river dolphin (Ala128Thr, 11.8Å to PSB; predicted blue-shift); and Weddell seal [(Thr311Ala, 9.3Å; predicted red-shift and Val/Met315Thr, 11.2; Å; predicted blue-shift (both to β-ionone ring)]. Interestingly, all melanopsins examined possess 126F (Fig 1) which is positioned just 5.0Å from the PSB (not shown), while the corresponding residue in vertebrate Rh1, Rh2 and LWS opsins is 94S/T and is highly conserved.

## Skyward solar radiance at dawn overlaps with Opn4 absorbance maxima

To better understand the biological relevance of the conserved absorbance maxima of vertebrate melanopsins, spectral radiance measurements were recorded to determine the light available to activate Opn4 expressing ipRGCs *in situ*. Spectral radiance measurements were taken in air and recorded between astronomical twilight (06:35 EDT) and sunrise (07:00 EDT). As shown in Fig 5, across timepoints, peak radiance values from the available light were slightly longer than 450 nm. Thus, the reported absorbance maximum of melanopsins of ~480 nm reasonably coincides with the most prevalent wavelengths of light recorded here.

**Table 1. Nonconserved amino acid substitutions in marine mammal melanopsins.**

| | Amino Acid Position and Distance to Chromophore | | | | | | | | | |
|---|---|---|---|---|---|---|---|---|---|---|
| Mouse | 89 | 93 | 123 | 128 | 187 | 233 | 295 | 311 | 315 | 342 |
| Squid | 54 | 58 | 88 | 94 | 153 | 199 | 263 | 279 | 283 | 310 |
| Distance (Å)[1] | 22.8 | 28.4 | 9.5 | 11.8 | 26.5* | 10.2* | 18.5* | 9.3* | 11.2* | 13.8** |
| Consensus | Thr | Thr | Ala | Ala | Ala | Ala | Glu | Thr | Val/Met | Tyr |
| Beluga whale | - | - | - | - | Thr | - | - | - | - | - |
| Harbor porpoise | - | - | - | - | Thr | - | - | - | - | - |
| Yangtze finless porpoise | - | - | - | - | Thr | - | - | - | - | - |
| Yangtze river dolphin | - | - | - | Thr | - | - | - | - | - | - |
| Weddell seal | - | - | - | - | - | - | - | Ala | Thr | His |
| Walrus | Met | - | - | - | - | - | - | - | Thr | His |
| Manatee | Met | Iso | Thr | - | - | Thr | Ala | - | - | His |
| Polar bear | Met | - | - | - | - | - | - | - | - | His |
| Mouse | - | - | - | - | - | - | Ala | - | - | His |

[1]Shortest distance to the chromophore measured in angstroms (Å) from amino acid functional group to the Schiff base nitrogen; the β-ionone ring* of the chromophore; or equal distance to the Schiff base nitrogen and β-ionone ring**.

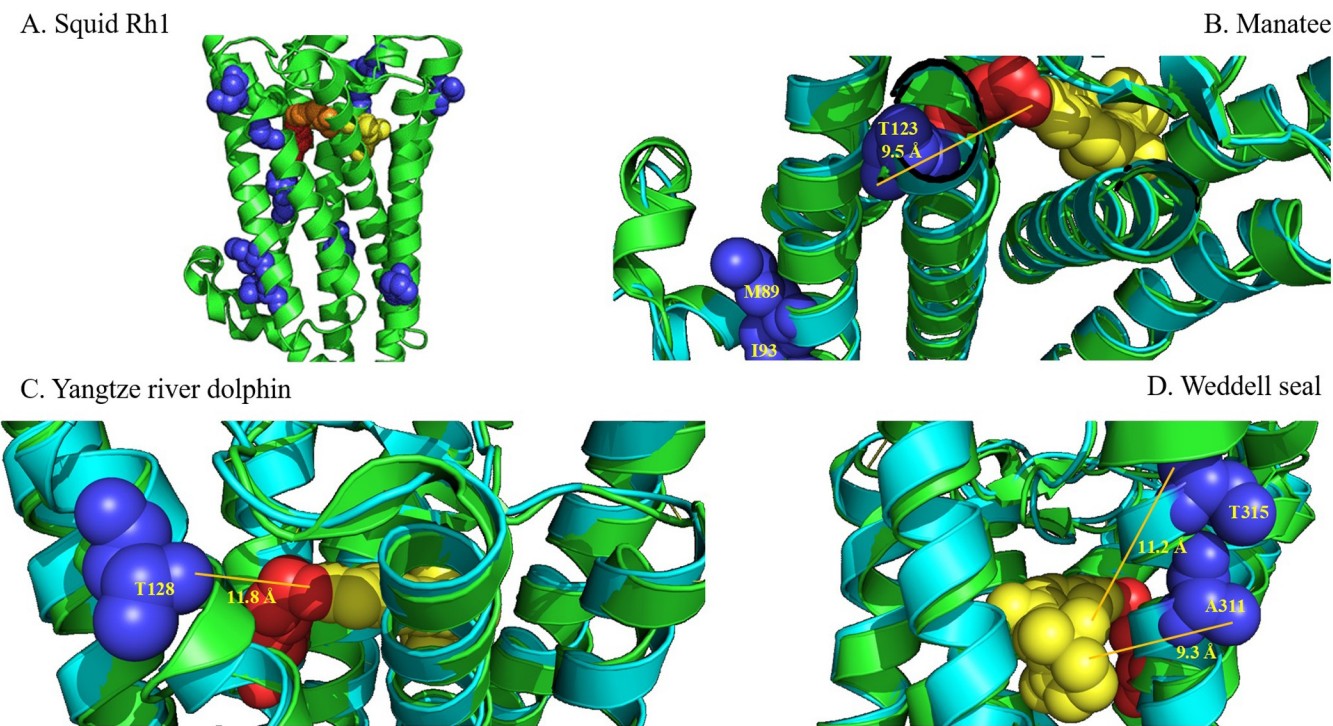

A. Squid Rh1

B. Manatee

C. Yangtze river dolphin

D. Weddell seal

**Fig 4. Structural modeling of marine mammal melanopsin (Opn4) spectral tuning amino acid positions.** 3-D images of marine mammal Opn4 proteins (teal) were aligned with squid Rh1 (green) to determine relative positions of helices and residues. (A) squid rhodopsin (Protein Data Bank accession number 2Z73) showing the relative position of the ten nonconservative amino acid substitutions from Fig 3. A proposed protonated Schiff base counterion (Glu214) [77–79] is shown in orange. Panels B-D highlight nonconservative amino acids potentially involved in spectral tuning and their relative predicted distance to the retinal Schiff base lysine (red) terminal NZ atom or the β-ionone ring of the chromophore (yellow): (B) manatee Thr123 (9.5 Å); (C) Yangtze river dolphin Thr128 (11.8 Å); and (D) Weddell seal Ala311 (9.3 Å); Weddell seal and walrus Thr315 (11.2 Å). Opn4 numbering is from mouse.

## Analysis of positive selection on marine mammal melanopsins

To investigate patterns of selection in marine mammal melanopsins, we used codon-aligned models to estimate the ratio of non-synonymous (amino-acid altering; Ka or dN) to synonymous (silent; Ks or dS) substitutions (Ka/Ks or dN/dS ratio, also referred to as ω) for a data set of the 17 marine mammal *Opn4* coding sequences shown in Fig 1. This was used to estimate both positive and purifying selection at each amino acid site. An alignment of melanopsin coding sequences was analyzed with Bayesian models which assume a statistical distribution to account for heterogenous ω values among sites. Rates were normalized to a value of 1 which result in a range of outcomes described as follows: neutral selection, ω = 1; negative or purifying selection, ω < 1; and positive selection, ω > 1. These values are shown in Fig 6 with a scalable selection ranging from 1 (strong positive selection) to 7 (strong purifying selection). It was hypothesized that sites resulting in positive selection result in the replacement of amino acids that were advantageous to the organism and fixed at a higher rate than that of neutral or synonymous mutations. As shown in Fig 6, the majority of sites throughout the coding sequence are shown to be under purifying selection and have been subjected to high level functional constraints during evolution. Purifying selection dominates within the seven transmembrane domains, while all putative spectral tuning positions describe above are under either weak purifying or neutral selection. In fact, the only positions that have undergone relatively strong positive selection are found in the N-terminal domain (Gln26, Arg38, Val44); the phosphorylation domain (Ser376); and in the C-terminal domain (Ser458, Pro464, Arg519).

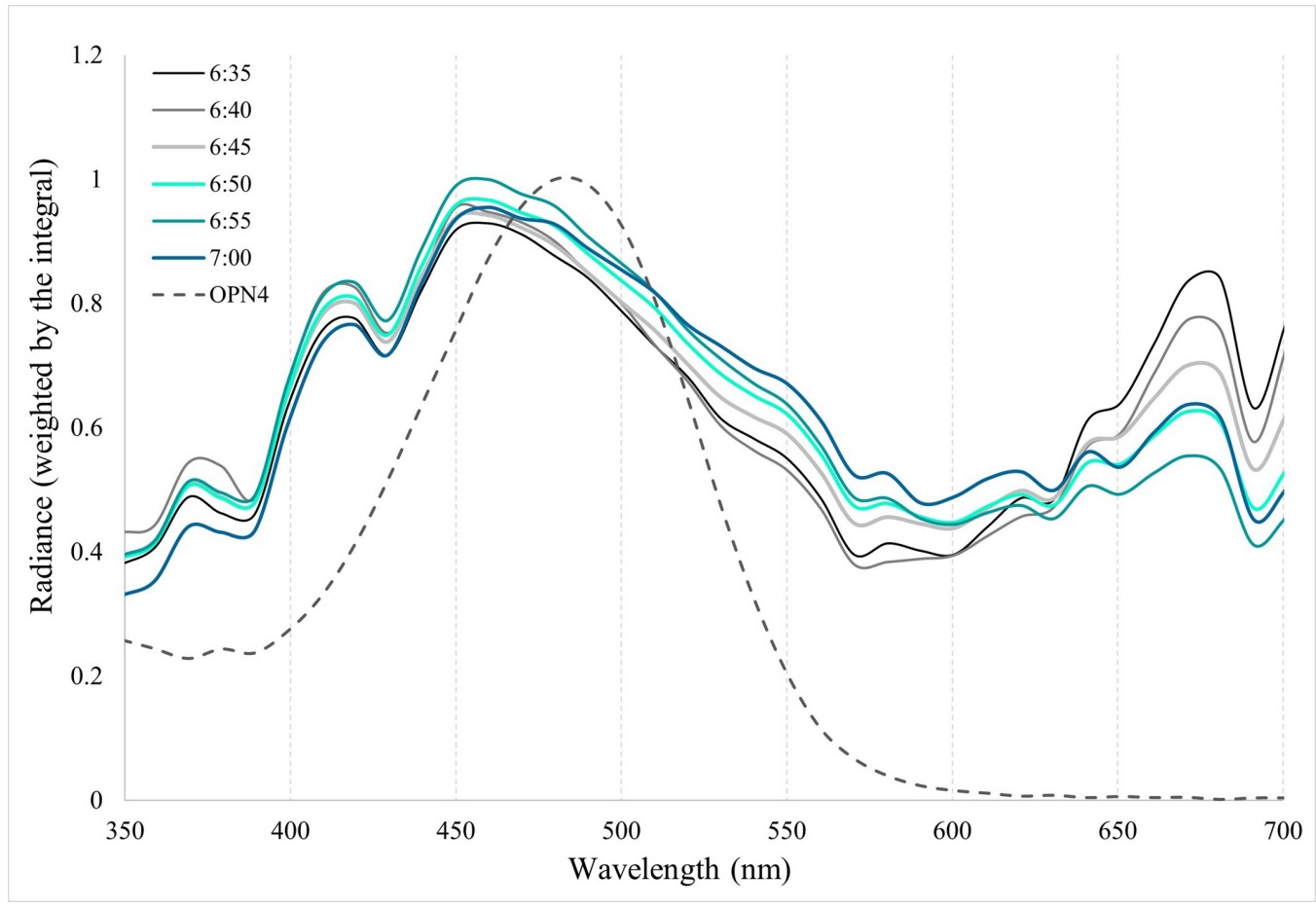

**Fig 5. Overlay of melanopsin (Opn4) absorbance spectrum with normalized solar spectral radiance values during sunrise.** A hypothetical $A_1$-chromophore-based Opn4 spectrum is plotted with absorbance maximum of 480 nm (dashed trace) with solar radiance spectra recorded at five-minute intervals from the onset of civil twilight (06:35 EDT) until just after sunrise (07:00 EDT). Spectral radiance measurements ($\mu W\ cm^{-2}\ nm^{-1}\ sr^{-1}$) are normalized by the integral (i.e., area under the curve) and scaled from zero to one.

Twelve putative phosphorylation sites consisting of conserved serines and threonines positioned in the carboxyl terminal domain (CTD), including Ser376 (see above), were examined for nonconservative amino acid substitutions (Fig 7). These sites have been previously identified as being responsible for the deactivation of the photoactivated melanopsin pigment [63, 64] and are separated into the PI and PII clusters of putative C-terminal phosphorylation sites. Naturally occurring amino acid substitutions at these phosphorylation sites in the Opn4 CTDs from seventeen marine mammals were identified by manual examination of amino acid alignments and compared to the phosphorylation sites found in mouse Opn4. As shown in Fig 7, there were amino acid substitutions at three potential phosphorylation sites found in both P1 and P2. Within P1, both the mysticete whales and polar bear possessed nonconserved Asn and Gly residues, respectively, at phosphorylation position 376, while all marine mammals examined possessed either Tyr or His at phosphorylation position 379. The Delphinidae and Phocoenidae species possessed Phe at position 384. Within P2, the Physeteroidea species possessed Asn at position 391, while all marine mammal species examined possessed Ala at position 394. The two Balaenidae species possessed Leu at position 395.

There are strongly conserved serine and threonine residues across all taxa at phosphorylation sites 372, 381, 385, 388, 389, and 392, all of which have undergone weak to strong

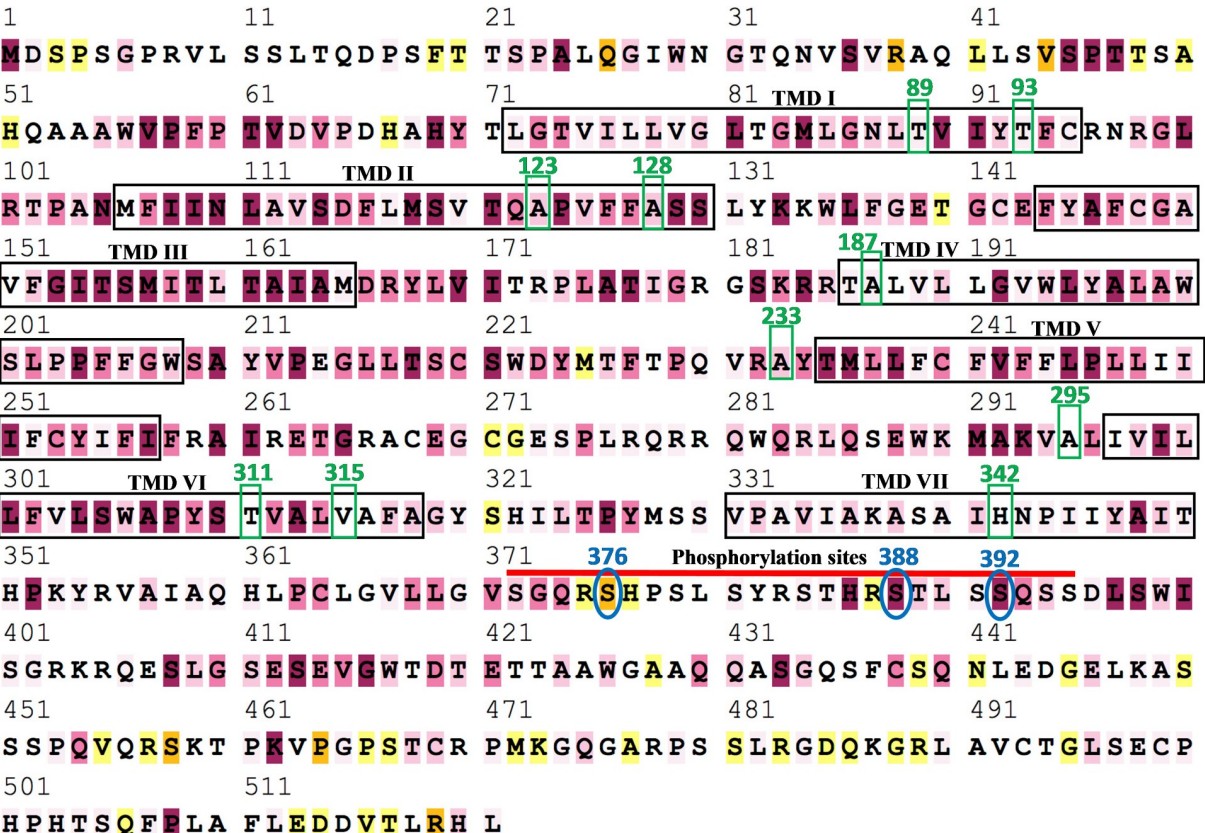

**Fig 6. Natural selection of marine mammal melanopsin (*Opn4*) amino acid sites.** Marine mammal *Opn4* amino acid sequences were aligned and the ratio of non-synonymous (amino-acid altering) to synonymous (silent) substitutions (Ka/Ks ratio) was used to estimate positive (orange), purifying (purple), or neutral (white) selection at each amino-acid site. Transmembrane domains are labeled and boxed. Putative spectral tuning positions are numbered by amino acid position and boxed in green. Phosphorylation sites (red bar) undergoing either strong positive or purifying selection are circled in blue. Mouse *Opn4* was used as template and for amino acid numbering (GenBank accession number NM_013887).

purifying selection (see Fig 6). Phosphorylation sites 384 and 391 have also undergone weak purifying selection but are not as well conserved in the marine mammals. Sites 379 and 394 have undergone either neutral or weak purifying selection and are well conserved in the marine mammals but lack either a Ser or Thr and are thus not phosphorylated. Site 395 is conserved in most marine mammals with the exceptions being the two Balaenidae species which possess Leu. All marine mammal melanopsins examined possess a Tyr at position 382, while the river dolphin possesses a Ser at this position. Interestingly, site 376 is the only putative phosphorylation site to have undergone strong positive selection with the baleen and toothed whales possessing Asn and Thr, respectively, at this position. The only other species possessing a non-phosphorable amino acid at this position is polar bear which possesses a Gly.

|  |  | PI |  |  |  |  |  |  |  |  |  |  | PII |  |  |  |  |  |
|---|---|---|---|---|---|---|---|---|---|---|---|---|---|---|---|---|---|---|
| **Amino Acid Position** | 372 |  |  |  |  |  |  |  | 380 |  |  |  |  | 385 |  |  |  | 390 | 395 |

| Species | 372 | 373 | 374 | 375 | 376 | 377 | 378 | 379 | 380 | 381 | 382 | 383 | 384 | 385 | 386 | 387 | 388 | 389 | 390 | 391 | 392 | 393 | 394 | 395 |
|---|---|---|---|---|---|---|---|---|---|---|---|---|---|---|---|---|---|---|---|---|---|---|---|---|
| Mouse | S | G | Q | R | S | H | P | S | L | S | Y | R | S | T | H | R | S | T | L | S | S | Q | S | S |
| Bowhead whale | . | . | . | H | N | G | L | Y | T | . | . | . | . | . | . | . | . | . | . | . | . | . | A | L |
| Right whale | . | . | . | . | N | G | L | Y | T | . | . | . | . | . | . | . | . | . | . | . | . | . | A | L |
| Minke whale | . | . | . | . | N | G | L | Y | T | . | . | . | . | . | . | . | . | . | . | . | . | . | A | . |
| Humpback whale | . | . | . | . | N | G | L | H | T | . | . | . | . | . | . | . | . | . | . | . | . | . | A | . |
| Sperm whale | . | . | . | . | T | G | L | Y | T | . | . | . | . | . | . | . | . | . | N | . | . | . | A | . |
| Dwarf sperm whale | . | . | . | . | T | G | L | Y | T | . | . | . | . | . | L | . | . | . | N | . | . | . | A | . |
| Bottlenose dolphin | . | . | . | . | T | G | L | Y | T | . | . | . | . | F | . | . | . | . | . | . | . | . | A | . |
| White-sided dolphin | . | . | . | . | T | G | L | Y | T | . | . | . | . | F | . | . | . | . | . | . | . | . | A | . |
| Killer whale | . | . | . | . | T | G | L | Y | T | . | . | . | . | F | . | . | . | . | . | . | . | . | A | . |
| Beluga whale | . | . | . | . | T | G | L | Y | T | . | . | . | . | F | . | . | . | . | . | . | . | . | A | . |
| Finless porpoise | . | . | . | . | T | G | L | Y | T | . | . | . | . | F | . | . | . | . | . | . | . | . | A | . |
| River dolphin | . | . | . | H | T | G | L | Y | T | . | S | . | . | F | . | . | . | . | . | . | . | . | A | . |
| Harbor porpoise | . | . | . | . | T | G | L | Y | T | . | . | . | . | F | . | . | . | . | . | . | . | . | A | . |
| Weddell seal | . | . | . | H | . | G | . | Y | T | . | . | . | . | . | . | . | . | . | . | . | . | . | A | . |
| Walrus | . | . | . | H | T | G | . | Y | T | . | . | . | . | . | . | . | . | . | . | . | . | . | A | . |
| Manatee | . | . | . | . | T | R | . | Y | T | . | . | . | . | . | . | . | H | . | . | . | . | . | A | . |
| Polar bear | . | . | . | . | G | G | . | Y | A | . | . | . | . | . | . | . | . | . | . | . | . | . | A | . |
| Cow | . | . | . | . | T | G | L | Y | T | . | . | . | . | . | . | . | . | . | . | . | . | . | A | . |

**Fig 7. Alignment of marine mammal melanopsin (Opn4) carboxyl terminal domains containing deactivating phosphorylation sites.** Putative phosphorylation sites identified in mouse [63, 64] are highlighted in red. Nonconservative amino acid substitutions of putative phosphorylation sites in marine mammal Opn4 opsins are highlighted in yellow. Ser382 in river dolphin is highlighted in red as a potential phosphorylation site. Bovine Opn4 amino acid substitutions are included as a reference.

## Opn4 activity based on fluorescent calcium imaging

The deactivation kinetics of Opn4 have been previously described in the mouse, along with the phosphorylation sites involved, and the resulting pupil light reflex (PLR) when these sites are mutated [63–66]. To better understand the PLR of marine mammals, specifically the cetacean rod monochromats, we examined the deactivation kinetics of the bowhead whale Opn4 pigment and the potential effect of the Asn376 substitution when compared to manatee and the mouse control, both of which are cone dichromats. Calcium imaging assays were performed that compared the deactivation rates of the bowhead whale Opn4 pigment with the deactivation rates from wild type mouse and manatee Opn4 pigments. A mouse Opn4 mutant, where all putative serine/threonine phosphorylation sites in the CTD were replaced with alanine residues (mouse phosphonull mutant: Pnull), was also used for comparison. The deactivation kinetics of the bowhead whale Opn4 pigment is significantly slower (p<0.0001) than both manatee and the mouse Opn4 control and is more similar to the deactivation rate of the Pnull pigment (Fig 8). While the deactivation rate of the manatee Opn4 pigment is also slower than the mouse Opn4 control (p<0.005), it is considerably faster than that of the bowhead whale Opn4 pigment.

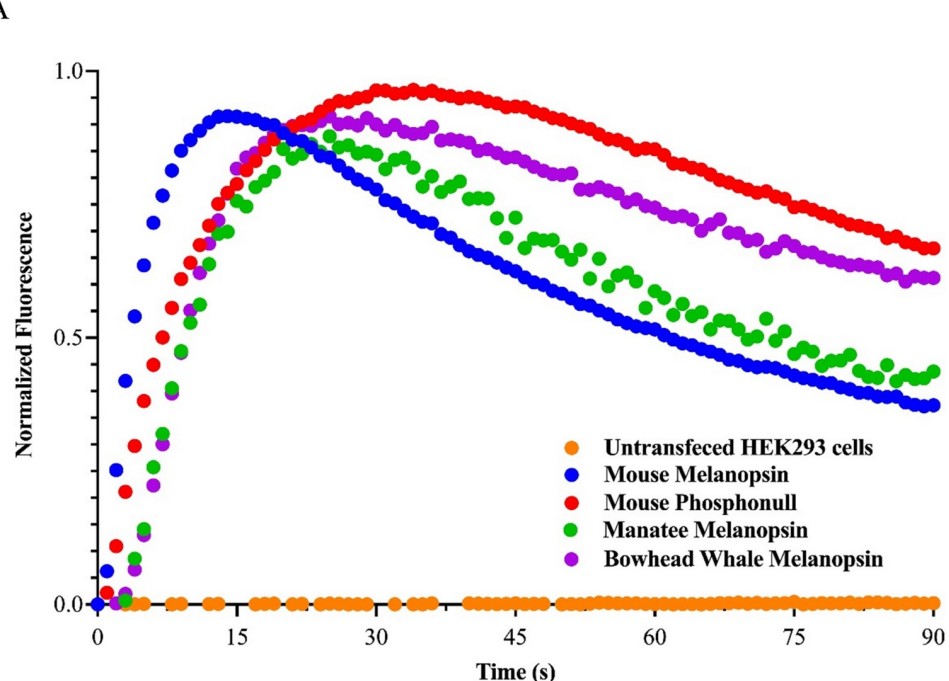

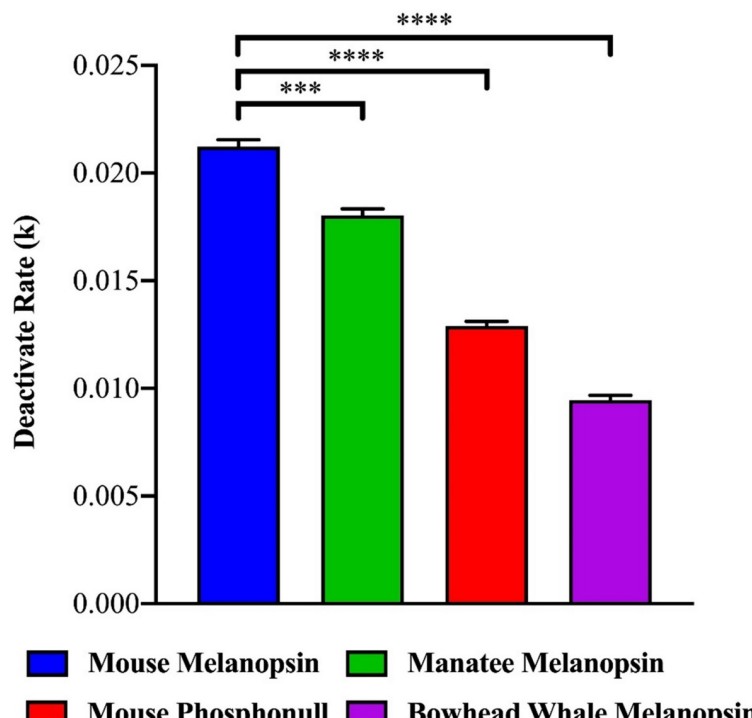

**Fig 8. Deactivation kinetics of melanopsin (Opn4) constructs.** A. *In vitro* Opn4 signaling kinetics based on normalized calcium imaging data of mouse Opn4 (blue trace), West Indian manatee (green trace), bowhead whale (purple trace), and mouse Opn4 lacking CTD phosphorylation sites (mouse phosphonull mutant—red trace). Negative control of untransfected HEK293 cells (orange trace) is shown for comparison. B. Deactivation rates are significantly different between mouse Opn4 and the other three constructs. The rate of deactivation of West Indian manatee Opn4

is more similar to mouse Opn4, while the rate of deactivation of bowhead whale Opn4 is more similar to mouse phosphonull mutant. All rates were compared to the mouse Opn4 deactivation rate. *** and **** correspond to p-values of <0.0005 and <0.0001, respectively. Error bars denote standard error of the mean.

## Discussion

This study addressed two principal questions pertaining to marine mammal melanopsins regarding the spectral tuning properties of the pigments and the role that melanopsin plays in the pupillary light reflex (PLR). With regards to the first question, we hypothesized that marine mammal melanopsins possess few if any amino acid substitutions that would result in a deviation from the absorbance maxima ($\lambda_{max}$) of typical vertebrate melanopsins (~480 nm) which appear to be spectrally tuned to the dominant solar spectral irradiance. To test this hypothesis, we examined *Opn4* sequences from a diverse set of marine mammal species and generated homology models from these sequences to identify amino acid substitutions that may play a role in altering the spectral tuning properties of the resulting pigments. While the results from this analysis identified a total of ten nonconserved substitutions across all species within or near the Opn4 transmembrane domains, homology modeling suggested that only five of these substitutions are within proximity to the chromophore to influence the electrostatic interaction at the protonated Schiff base (PSB) of the chromophore. Residues Thr123 in manatee and Thr128 in Yangtze river dolphin both introduce an OH-group proximal to the PSB of the chromophore when compared to the consensus sequence. These two amino acids have predicted distances of less than 12 Å to the PSB of their respective chromophores and may be close enough to influence the spectral tuning of each pigment. Applying the OH-site rule [57], we predict that both substitutions would result in slightly blue-shifted absorbance maxima when compared to the other mammalian Opn4 pigments examined. If substitutions in the extracellular loop domains are included, the Thr233 substitution in manatee Opn4, which is approximately 10 Å from the β-ionone ring of the chromophore, would most likely result in a slightly red-shifted absorbance maximum thus negating the contribution from the Thr123 substitution in this pigment. The Weddell seal Opn4 pigment possesses two amino acid substitutions (Ala311 and Thr315), while walrus Opn4 possesses a single substitution (Thr315), which are all positioned less than 12 Å from the β-ionone ring of the chromophore. The substitutions in Weddell seal are predicted to result in slight red- and blue-shifts, respectively, potentially canceling any spectral movement relative to 480 nm, while the single substitution in walrus is predicted to result in a slightly red-shifted absorbance maximum. As noted in the Results section, all vertebrate melanopsins examined possess Phe126 which is positioned very near the PSB (~5.0Å). Interestingly, the corresponding residues in vertebrate Rh1, Rh2 and LWS opsins are highly conserved and hydroxyl-bearing (e.g., Ser and Thr), while highly conserved nonpolar residues (e.g., Ala and Val) are observed in vertebrate SWS-1 and SWS-2 opsins. Fasick *et al*. (2002, [41]) showed dramatic spectral shifts in the SWS-1 pigments associated with amino acid position 86 (bovine Rh1 numbering) which is approximately one helical turn away from position 126 discussed here, while also sharing a similar distance to the PSB of the chromophore (~5.0Å) and similar amino acid substitutions (Phe and Tyr). Thus, position 126 may influence the spectral shift of melanopsins into the blue region of the spectrum. Due its close proximity to the PSB of the chromophore, position 126 and its associated amino acids may be of interest in future mutagenesis experiments examining the spectral tuning properties of melanopsins.

When considering the position and nature of the amino acid substitutions in marine mammal melanopsins discussed here, we predict that the absorbance spectra and maxima of marine mammal melanopsins should be similar to those described and experimentally determined

from other vertebrate melanopsins ($\lambda_{max}$ ~480 nm; [52–56]). This conclusion is based on the conservation of the vast majority of residues found throughout the transmembrane domains of vertebrate melanopsins and the strong purifying selection placed upon these residues as shown in Fig 6. However, functional confirmation of this claim requires suitable cell culture and purification systems to provide *in vitro* expression of the dark-adapted absorbance spectra of these pigments that can then be compared to the values reported from *in vivo* recordings from model organisms.

To date, there is no clear and simple explanation as to why melanopsins are spectrally tuned to 480 nm light. The conserved nature of melanopsin absorbance maxima across taxa is most likely the result from selection pressures on these pigments to maximally absorb the dominant short-wavelength spectral irradiance associated with dawn and dusk, illuminating both terrestrial environments and surface water in a similar fashion [80, 81]. However, the dominant wavelengths of solar spectral irradiance present in surface water environments is around 510 nm [74, 82]. Our measurements of spectral radiances at dawn, as shown in Fig 5, have peaks closer to 450 nm. One conclusion from this apparent mismatch between the absorbance maximum of the pigment and the transmittance maxima of ambient natural light is that melanopsins have undergone selection pressures that have positioned the absorbance maximum of the pigment between these two transmittance peaks at dawn (~450 nm) and during the day (~510 nm) in order to maximize photon capture throughout the day from dawn to dusk. Perhaps coincidental, the average of these two peak transmittance values is 480 nm.

The second question addressed in this study pertains to the role of melanopsin expressed in intrinsically photosensitive retinal ganglion cells (ipRGCs) and the pupillary light reflex, with attention given to the cetacean rod monochromats. Rod photoreceptors are responsible for directing the PLR under dim light (scotopic/mesopic) conditions in all mammals [61, 83]. With this understood, it could then be supposed that the ipRGCs expressing melanopsin in mammalian rod monochromats would assume the role of the missing cone photoreceptors in bright light (photopic) conditions to prevent rod photoreceptor bleaching. However, this is not likely the case as cone input to ipRGCs is relatively weak and that ipRGCs expressing melanopsin are the dominant determinant of pupil size during the day [61]. In addition to the PLR, anatomical features have evolved to reduce the amount of light entering the eyes of aquatic organisms including marine mammals. Cetaceans possess dorsal operculum pupillarae, that when fully constricted result in a double-slit pupil [84] that focuses light onto the two best vision areas of the retina [85]. In the case of an aquatic mammal, a pupil design like this may function to reduce the intensity of the brighter downwelling light, while leaving the dimmer sidewelling light less attenuated and available for visual processes [86]. We hypothesized that, in addition to a double-slit pupil, it would be advantageous in cetacean rod monochromats to extend the length of time of pupillary constriction when transitioning rapidly between photopic and scotopic light environments, as is the case during traveling behaviors, to prevent photobleaching of the all-rod retina upon returning to the surface to breathe. To test this hypothesis, we examined the deactivation kinetics of melanopsin from two marine mammal species, a rod monochromat: bowhead whale (*Balaena mysticetus*); and a cone dichromat: West Indian manatee (*Trichechus manatus*), with mouse (*Mus musculus*), also a cone dichromat, as control. While the deactivation kinetics for bowhead whale melanopsin was previously reported in a comparison to that of a whale shark (*Rhincodon typus*)–mouse (*Mus musculus*) chimeric melanopsin [87], the deactivation kinetics experiments with bowhead whale melanopsin were repeated here to compare deactivation rates from a rod monochromat with those from a marine mammal possessing a duplex retina and dichromatic color vision. Our results show a slower rate of

deactivation of melanopsin from bowhead whale when compared to the deactivation rates from either manatee or mouse wildtype. With regards to bowhead whale deactivation rates, this result is most likely due to the combination of four nonconservative amino acid substitutions in the phosphorylation domain at positions 376, 379, 394, and 395 with 376 and 395 being exclusive to the Balaenidae species. The deactivation rate of manatee melanopsin was also significantly slower compared to the mouse wildtype control. This result is most likely due to nonconservative amino acid substitution at phosphorylation sites 379 (Ser379Tyr) and 394 (Ser394Ala) which are shared by all of the mammals examined, including cow (*Bos taurus*), but not mouse. It is not entirely clear what selection pressures are exerted on positions 379 and 394 in mammalian melanopsins. When the phosphorylation domains are examined from examples of both strictly nocturnal (e.g., small-eared galago, *Otolemur garnettii*) and strictly diurnal (e.g., thirteen-lined ground squirrel, *Ictidomys tridecemlineatus*) mammals, both species possess Ala394 like the manatee (Accession numbers found in S4 Table). When these same sequences are examined at position 379, the nocturnal galago possesses Tyr379 like the marine mammals and diurnal cow, while the diurnal squirrel possesses Ser379 like the nocturnal mouse. In this study it was shown that both positions 379 and 394 are under neutral or slight purifying selection, respectively, unlike position 376 which is under strong positive selection. This suggest that positions 379 and 394 may not serve a critical role in deactivation kinetics as do the other phosphorylation sites.

In summary, the results presented here suggest that melanopsins from 17 marine mammal species that inhabit diverse spectral environments maintain a highly conserved amino acid consensus sequence with their terrestrial counterparts with regards to the protein domains responsible for the pigment's absorbance spectrum and absorbance maximum. This level of amino acid conservation, however, is not observed in the phosphorylation domain of the carboxyl tail where all cetaceans examined possessed fewer phosphorylation sites when compared to a terrestrial control (mouse) sequence resulting in significantly slower deactivation kinetics in the two marine mammal melanopsin pigments examined by calcium imaging. To conclude, we suggest that the strongly conserved residues within the melanopsin chromophore binding pocket are maintained to absorb spectral solar radiance in the same region of the visible spectrum for both aquatic and terrestrial mammals alike. In contrast, the phosphorylation sites in marine mammal melanopsins responsible for pigment deactivation diverge when comparing species possessing duplex retinae containing both functional rod and cone photoreceptors with rod monochromats lacking functional cones, with the latter possessing significantly longer melanopsin deactivation rates relative to the former.

## Supporting information

**S1 Table. Oligonucleotide primers for PCR.** Forward and reverse primers for PCR amplification of full-length bowhead whale Opn4 and partial coding domain for other cetacean Opn4 sequences including North Atlantic right whale, bottlenose dolphin and harbor porpoise. (DOCX)

**S2 Table. PCR cycling parameters.** Temperatures for denaturation, annealing and extension for the amplification of Opn4 coding sequences from bottlenose dolphin, harbor porpoise and North Atlantic right whale retinal cDNA. Reactions contained forward and reverse primers (see above) and were carried out in a 25 μl mixture containing 1 μM of each primer, 2 μl of reverse transcribed single-stranded cDNA, and Amplitaq Gold master mix (Invitrogen, Carlsbad, CA). Annealing Tm were from NCBI Primer-BLAST for each primer pair. (DOCX)

**S3 Table. PCR cycling parameters.** Temperatures for denaturation, annealing and extension for the amplification of West Indian manatee and bowhead whale opn4 coding sequences. (DOCX)

**S4 Table. Accession numbers.** GenBank accession numbers used to generate the tree shown in Fig 2 and in commented on in the Discussion. (DOCX)

**S1 Text.** (TXT)

## Acknowledgments

We thank William McLellan, University of North Carolina-Wilmington for the collection of the North Atlantic right whale eye and Kristi West, Hawaii Pacific University, for the collection of the humpback and dwarf sperm whale eyes. All tissues were received by J.I.F. under a letter of authorization granted by the National Marine Fisheries Service Northeast Regional Office. We thank R. Ewald, University of Tampa, and V. Ortiz and A. Caswell, Kean University, for their participation on this project. This manuscript is dedicated to the memory of our colleague and friend, Dr. Sherri Ann (Goldberg) Eldridge (Woods Hole Oceanographic Institution).

## Author Contributions

**Conceptualization:** Jeffry I. Fasick, Phyllis R. Robinson.

**Data curation:** Jeffry I. Fasick, Haya Algrain, Courtland Samuels, Padmanabhan Mahadevan, Lorian E. Schweikert, Zaid J. Naffaa.

**Formal analysis:** Jeffry I. Fasick, Haya Algrain, Courtland Samuels, Padmanabhan Mahadevan, Lorian E. Schweikert, Phyllis R. Robinson.

**Funding acquisition:** Jeffry I. Fasick, Phyllis R. Robinson.

**Investigation:** Jeffry I. Fasick, Padmanabhan Mahadevan, Lorian E. Schweikert, Zaid J. Naffaa, Phyllis R. Robinson.

**Methodology:** Jeffry I. Fasick, Padmanabhan Mahadevan, Lorian E. Schweikert, Phyllis R. Robinson.

**Project administration:** Jeffry I. Fasick.

**Resources:** Jeffry I. Fasick, Phyllis R. Robinson.

**Supervision:** Jeffry I. Fasick, Phyllis R. Robinson.

**Writing – original draft:** Jeffry I. Fasick.

**Writing – review & editing:** Jeffry I. Fasick, Haya Algrain, Courtland Samuels, Padmanabhan Mahadevan, Lorian E. Schweikert, Phyllis R. Robinson.

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
