## [Decision Letter · Decision Letter 0]

22 Jul 2021

PONE-D-21-18150

Spectral tuning and deactivation kinetics of marine mammal melanopsins

PLOS ONE

Dear Dr. Fasick,

Thank you for submitting your manuscript to PLOS ONE. After careful consideration, we feel that it has merit but does not fully meet PLOS ONE’s publication criteria as it currently stands. Therefore, we invite you to submit a revised version of the manuscript that addresses the points raised during the review process.

As you will see from the attached reviewer comments, the required changes are of editorial nature only.

We look forward to receiving your revised manuscript.

Kind regards,

Stephan C.F. Neuhauss, Ph.D.

Academic Editor

PLOS ONE

Reviewers' comments:

Reviewer's Responses to Questions

**Comments to the Author**

1. Is the manuscript technically sound, and do the data support the conclusions?

Reviewer #1: Yes

Reviewer #2: Yes

2. Has the statistical analysis been performed appropriately and rigorously? 

Reviewer #1: Yes

Reviewer #2: Yes

3. Have the authors made all data underlying the findings in their manuscript fully available?

Reviewer #1: Yes

Reviewer #2: Yes

4. Is the manuscript presented in an intelligible fashion and written in standard English?

Reviewer #1: Yes

Reviewer #2: Yes

5. Review Comments to the Author

Reviewer #1: Fasick et al. examined melanopsin (Opn4) in marine mammals. Melanopsin is a recently discovered photopigment expressed, in mammals, in a subset of retinal ganglion cells. It renders these cells intrinsically photosensitive (ipRGCs). ipRGCs play a major role in “non-visual” responses to light, a set of functions that includes photoentrainment of the circadian clock and pupillary reflex to light (PLR).

Marine mammals’ visual system, in particular the photopigments and photoreceptors, adapted to marine habitat. Melanopsin has never been studied in marine mammals. Here, the authors question if, similarly to what was observed for other photopigments, marine mammal’s melanopsin has undergone adaptations to the marine environment. Specifically, they ask if melanopsin protein domains related to spectral sensitivity and deactivation kinetics diverged from their terrestrial counterparts.

Based on homology modeling of the marine mammal Opn4 sequences, they first look for residues that may interfere with the domains critical for the absorption spectrum of melanopsin. They detected 10 non-conserved residues in positions potentially impacting spectral tuning, of which very few were in range (in only 3/17 species) which suggests that the melanopsin peak may not be shifted in marine mammals. Consistent with this result, they found little evolutionary constraint on these residues compared to other residues of the transmembranary domains.

On the contrary, the sites of several phosphorylable residues of the C terminus tail, important for deactivation speed in the mouse, were subject to strong selection. Accordingly, responses to light from marine melanopsins (heterologously expressed in HEK cells) were slower to stop. The authors hypothesize that a longer melanopsin response, if it translates to melanopsin-dependent responses such as PLR, may be beneficial to rod regeneration in rod monochromate animals.

While some conclusions of this study rest on the assumption that the modelization and the in vitro responses translate to the physiology of these animals, this is an intriguing comparative study that will be of interest to a very large range of biologists.

I do not have any major reservations. I just listed below a few points that, I think, would deserve to be clarified.

L133. Refs?

L144. The authors do not explicitly mention the phylogenic proximity of some species with the cow used as a reference. It may not be obvious to the readership

L223. What was the irradiance of the flashes?

L373. Spectral measurements were performed “between astronomical twilight (06:35 EDT) and sunrise (07:00 EDT)”. What was the reasoning behind performing the spectral measurements only at dawn? Also, the readership would benefit from some descriptions of the marine species habitat/habits. For example: What is the amount and schedule of light sampling at the surface in the marine mammals considered? That, given the vast diversity of species considered, may be quite different. What is the underwater spectrum at the depth where these species are living?

L389. (Ka/Ks or dN/dS ratio, also referred to as ω), not clear.

L455. The authors may also want to cite Mure et al., 2016 regarding the impact of Opn4 C-terminus tail phosphorylation on the PLR.

L455. Somasundaram et al., 2017 is referenced twice (69 and 83).

L525. “functional confirmation of this claim requires suitable cell culture and purification systems to provide in vitro expression of the dark-adapted absorbance”. While I agree with that, one cannot help but wonder why the authors made no attempt of spectral measurements with the calcium assay they used to compare deactivation kinetics?

Regarding absorption peak, the authors could discuss fish melanopsins as fishes share the same habitat but do not surface. Are some fishes’ melanopsin spectra known (and shifted?)?

Reviewer #2: The study by Fasick and colleagues investigate melanopsin (encoded by Opn4) in marine mammals. This is a most interesting research question, due to the high interest that this photopigment enjoys. Melanopsin is the photopigment of intrinsically photosensitive ganglion cells. These cells play a crucial role in the vertebrate retina and are fascinating for multiple reasons, including their rhabdomer-like visual transduction.

Somewhat surprisingly, this gene has not be studied in marine mammals, despite the large attention it has enjoyed in terrestrial mammals. This study now sets out to close this gap and to analyze potential adaptations to the marine environment.

The study is based on bioinformatic and in vitro approaches, which is a great starting point.

The main findings include the (to this reviewer) surprising conclusion that there was little evidence for spectral tuning. On the other hand, residues at the C-terminal, known to be subject to phosphorylation, that triggers the deactivation cascade. This conclusion was confirmed by in vitro experiments in HEK cells, where marine mammalian melanopsin was indeed slower in their shut-off kinetics.

Overall this is an interesting study that will be of interest to a diverse group of biologists. This study should also inspire other scientists to test these properties in vivo (not easily accessible of course).

I only have a few minor points that the authors should address:

1. The connection to the pupillary light response needs more supportive references

2. In general the authors should comment more on the visual ecology of these species. This includes some information on the spectral composition of their light environment and the illumination levels.

3. The discussion would be even more interesting by a comparison of other marine life, e.g. what is known about fish melanopsins, are there difference between surface and deep sea fish … But this is of course at the discretion of the authors.

Otherwise this is a most interesting and well written manuscript that will find a wide readership.

6. PLOS authors have the option to publish the peer review history of their article (what does this mean?). If published, this will include your full peer review and any attached files.

Reviewer #1: No

Reviewer #2: **Yes: **Jingjing Zang

---

## [Author Response · Author response to Decision Letter 0]

27 Aug 2021

August 27, 2021

Dear Dr. Neuhauss, 

We the authors have revised our manuscript (Manuscript ID PONE-D-21-18150) entitled, Spectral Tuning and Deactivation Kinetics of Marine Mammal Melanopsins for review by you as well as by the referees who enthusiastically agreed to review the revised manuscript. We have revised the manuscript by addressing the comments and concerns of the Academic Editor, and referees #1 and #2. The comments and concerns of the Academic Editor and the referees were extremely helpful in crafting a more sophisticated and concise manuscript and we thank the Academic Editor and referees for their time and well-thought-out critiques. Please note that all line numbers (denoted L) mentioned in this rebuttal refer to the line numbers from the original manuscript (file name: Manuscript) as do the reference numbers.

Comments and Responses

Academic Editor

Comment 1. Data available upon request. When filling out the submission for review, this option was a little confusing. Our data (DNA sequences) will be available and uploaded to NCBI when the manuscript is accepted for publication. Sequence accession numbers will be provided when available.

Reviewer #1

Comment 1. L133. Refs? The sentence ending on L133 proposes two hypotheses pertaining to mechanisms that may counter photoreceptor bleaching in bright light conditions. To our knowledge, there has been no research on the selection for rod visual pigments or mutants that reconstitute chromophore significantly faster. Our current work, of course, pertains to the second hypothesis proposed on L133, a compensatory mechanism to maintain an extended PLR wherein the pupil dilates considerably slower. 

Comment 2. L144. The authors do not explicitly mention the phylogenic proximity of some species with the cow used as a reference. It may not be obvious to the readership. We agree with Reviewer #1’s comment as this phylogenic proximity was buried at the end of Figure legend 2. Even-toed ungulates, like cow, and cetaceans belong to the same order, Cetartiodactyla. We have edited the Materials and Methods as follows: “Domestic cow Opn4 (Bos Taurus, GenBank accession no. NM_001192399) was used as a query sequence in nucleotide blast analyses to identify orthologs from other Cetartiodactyla marine mammal genomes including…”

Comment 3. L223. What was the irradiance of the flashes? Irradiance values from the Xenon flash lamp were not measured nor listed in the specifications of the instrument. However, a 5W Xenon flash lamp typically has irradiance values between 5-15 mWcm2.

Comment 4. L373. Spectral measurements were performed “between astronomical twilight (06:35 EDT) and sunrise (07:00 EDT)”. What was the reasoning behind performing the spectral measurements only at dawn? Also, the readership would benefit from some descriptions of the marine species habitat/habits. For example: What is the amount and schedule of light sampling at the surface in the marine mammals considered? That, given the vast diversity of species considered, may be quite different. What is the underwater spectrum at the depth where these species are living? With regards to the first question (…why sample at dawn?), this was simply the only time available to us on that day. At the time, JIF was studying photoentrainment pertaining to circadian rhythms. In retrospect, we now would have preferred to have sampled at high noon as well as at dusk to have a full account of the solar spectral measurements. With regards to the second question (What is the amount and schedule of light sampling at the surface in the marine mammals considered?), this may be found in L533-555 where we state, “However, the dominant wavelengths of solar spectral irradiance present in surface water environments is around 510 nm [77, 86]”, with radiance values near 1 Lw (found in reference 86). With regards to the third comment (What is the underwater spectrum at the depth where these species are living?), we address this along with Reviewer #2’s comment (“…some information on the spectral composition of their light environment and the illumination levels”). Readers should be able to use the references found in lines 92-102 with regards to visual foraging ecology of marine mammals: “…marine mammal visual pigments are spectrally tuned to overlap the underwater spectral radiance associated with foraging depth [9, 12, 42-51]. Modulating the spectral sensitivity results in relatively large, blue-shifted absorbance spectra from visual pigments of deep-diving pelagic marine mammals, and relatively slightly red-shifted absorbance spectra from the visual pigments of near-coastal and riverine species, when compared to their terrestrial counterparts. Unlike terrestrial mammals, most marine mammal species reside in two spectrally distinct photic environments: the surface where they breathe and utilize broadband light spectra, and at foraging depth where they can utilize narrowband light spectra. The spectral irradiance at foraging depth is typically blue-shifted from the 500 nm region of the visible light spectrum where terrestrial Rh1 pigments maximally absorb [7, 9, 44, 45, 57-60]”, with reference 58 providing radiance values as a function of depth for the North Atlantic right whale. Outside of this, describing the underwater spectral qualities of light for each species is beyond the scope of this paper. 

Comment 5. L389. (Ka/Ks or dN/dS ratio, also referred to as ω), not clear. We agree with Reviewer #1’s comment and have edited this sentence as follows: “To investigate patterns of selection in marine mammal melanopsins, we used codon-aligned models to estimate the ratio of non-synonymous (amino-acid altering; Ka or dN) to synonymous (silent; Ks or dS) substitutions (Ka/Ks or dN/dS ratio, also referred to as ω)…”

Comment 6. L455. The authors may also want to cite Mure et al., 2016 regarding the impact of Opn4 C-terminus tail phosphorylation on the PLR. We agree with Reviewer #1 and have added this reference.

Comment 7. L455. Somasundaram et al., 2017 is referenced twice (69 and 83). This has been corrected.

Comment 8. L525. “why have the authors made no attempt of spectral measurements with the calcium assay they used to compare deactivation kinetics? The authors agree and this is an experiment that will be performed in the future. It requires a modification of the instrument such that single monochromatic light is flashed to a select number of samples.

Comment 9. Regarding absorption peak, the authors could discuss fish melanopsins as fishes share the same habitat but do not surface. Are some fishes’ melanopsin spectra known (and shifted?)? We agree with Reviewer #2 that this would be an interesting comparison and we hope to begin similar studies with marine fish. An article that interested us examined melatonin inhibition in damselfish in the presence of blue light as opposed to green or red light (Takeuchi et al. 2014. Gen. Comp. Endocrin. 204: 158-165) but did not examine the damselfish melanopsin gene or its expression. To our knowledge, the spectral tuning properties in association with calcium imaging assays of marine fish melanopsins have not been done. 

Reviewer #2

Comment 1. The connection to the pupillary light response needs more supportive references. It is not entirely clear to us what Reviewer #2 is referring to with regards to “connection”? We can only assume that Reviewer #2 is referring to the “connection” between light, melanopsin expressing ipRGCs, as well as the innervation of rod and cone photoreceptors on these ipRGCs. The following 5 references directly examine the pupillary light response with regards to Opn4 expressing ipRGCs, while several other references in the manuscript describe the roles of rod and cone photoreceptors. We believe that we have included the appropriate references in the manuscript. We included a new reference per the request of Reviewer #1 (Mure et al., 2016).

Markwell EL, Feigl B, Zele AJ. Intrinsically photosensitive melanopsin retinal ganglion cell contributions to the pupillary light reflex and circadian rhythm. Clinic Exp Optomet. 2010;93(3):137-49. https://doi.org/10.1111/j.1444-0938.2010.00479.x

Keenan WT, Rupp AC, Ross RA, Somasundaram P, Hiriyanna S, Wu Z, et al. A visual circuit uses complementary mechanisms to support transient and sustained pupil constriction. eLife 2016;5: e15392 doi: 10.7554/eLife.15392

Lee SK, Sonoda T, Schmidt TM. M1 Intrinsically Photosensitive Retinal Ganglion Cells Integrate Rod and Melanopsin Inputs to Signal in Low Light. Cell Reports. 2019;29(11):3349-55. e2. https://doi.org/10.1016/j.celrep.2019.11.024

Mure LS, Hatori M, Zhu Q, Demas J, Kim IM, Nayak SK, et al. Melanopsin-encoded response properties of intrinsically photosensitive retinal ganglion cells. Neuron. 2016; 90: 1016–27. http://dx.doi.org/10.1016/j.neuron.2016.04.016

Somasundaram P, Wyrick GR, Fernandez DC, Ghahari A, Pinhal CM, Richardson MS, et al. C-terminal phosphorylation regulates the kinetics of a subset of melanopsin-mediated behaviors in mice. PNAS USA. 2017:201611893. https://doi.org/10.1073/pnas.1611893114

Comment 2. In general, the authors should comment more on the visual ecology of these species. This includes some information on the spectral composition of their light environment and the illumination levels. See Reviewer #1, Comment 4.

Comment 3. The discussion would be even more interesting by a comparison of other marine life, e.g., what is known about fish melanopsins, are there difference between surface and deep-sea fish … But this is of course at the discretion of the authors. See Reviewer #1, Comment 9.

Thank you for your consideration and please contact me for any further comments and corrections,

Sincerely,

Jeffry I Fasick, PhD

---

## [Editor Report · Decision Letter 1]

1 Sep 2021

Spectral tuning and deactivation kinetics of marine mammal melanopsins

PONE-D-21-18150R1

Dear Dr. Fasick,

We’re pleased to inform you that your manuscript has been judged scientifically suitable for publication and will be formally accepted for publication once it meets all outstanding technical requirements. Congratulations to a most interesting study.

Kind regards,

Stephan C.F. Neuhauss, Ph.D.

Academic Editor

PLOS ONE

---

## [Editor Report · Acceptance letter]

8 Oct 2021

PONE-D-21-18150R1 

Spectral tuning and deactivation kinetics of marine mammal melanopsins 

Dear Dr. Fasick:

I'm pleased to inform you that your manuscript has been deemed suitable for publication in PLOS ONE. Congratulations! Your manuscript is now with our production department. 

Kind regards, 

on behalf of

Dr. Stephan C.F. Neuhauss 

Academic Editor

PLOS ONE